# Effects of Straw Decomposition on Soil Surface Evaporation Resistance and Evaporation Simulation

**DOI:** 10.3390/plants14030434

**Published:** 2025-02-02

**Authors:** Shengfeng Wang, Longwei Lei, Yang Gao, Enlai Zhan

**Affiliations:** 1School of Water Conservancy, North China University of Water Resources and Electric Power, Zhengzhou 450046, China; 15237624073@163.com (L.L.); holmelai@163.com (E.Z.); 2Institute of Farmland Irrigation, Chinese Academy of Agricultural Sciences, Xinxiang 453002, China; gaoyang@caas.cn

**Keywords:** evapotranspiration, soil water content, model simulation, water-saving mechanism, straw incorporation, soil moisture retention curve

## Abstract

As a prominent agricultural country, China has widely implemented returning straw to the field in agricultural production. However, as the decomposition of straw progresses, the physical properties of the soil change, inevitably leading to alterations in the soil surface evaporation model. This study investigated the variations in soil evaporation rate, soil moisture content over 60 days after returning straw to the field, and bare soil through two leaching pond experiments. Through soil moisture retention curves at different degrees of decomposition, this study analyzed the impact of straw decomposition on soil’s water retention capacity. Based on measured data, this study formulated models for the soil surface evaporation resistance of bare soil and varying degrees of straw decomposition. With the comparison and contrast between the models, this study clarified the impact of straw decomposition on soil surface evaporation resistance. The main conclusions are the following: The moisture content of the surface soil decreases exponentially over time and, after 40 days of straw decomposition, the water content of the soil under decomposition is higher than that of bare soil. As the moisture content decreases, the cumulative evaporation from the soil increases linearly. The cumulative evaporation of the decomposed straw soil is lower than that of bare soil, with a relative reduction ranging from 3.08% to 32.2%. The straw decomposition significantly enhances the water retention capacity of the soil in the medium-to-high suction range. The straw decomposition enhances the evaporation resistance of the soil surface, and the greater the degree of decomposition, the more significant the enhancement effect. The research findings not only provide a scientific basis for agricultural water management, but also possess practical implications for guiding farmers to adopt more effective moisture retention measures.

## 1. Introduction

Soil evaporation, the process by which water from the soil surface transfers into the atmosphere in the state of water vapor, is the main link of farmland water circulation [1,2,3]. It not only dominates the water–heat balance of farmland, but also directly affects crop growth and water management [4,5].

As a complicated process, soil evaporation can be affected by temperature, humidity, wind speed, and other natural conditions [6,7,8,9]. Many scholars have conducted extensive research on the calculation of soil evaporation and put forward a series of theoretical and empirical calculation formulas [10,11,12,13,14,15]. Van Bavel and Hillel [15] introduced a theoretical method to calculate the aerodynamic resistance of soil evaporation. Yang and Blackewll et al. [12] proposed the one-layer and two-layer theoretical models of soil surface evaporation resistance. Yamanaka [13] presented the surface evaporation resistance model based on the surface energy balance, and verified it using the wind tunnel test.

In field irrigation, soil evaporation is generally regarded as an unproductive form of water loss. Hence, reducing this part of water consumption is crucial for the improvement of water use efficiency and development of efficient farmland water management measures [16,17]. There is no denying that measures should be taken to inhibit soil evaporation and promote agricultural production, such as applying mulch to the soil surface. Many scholars have studied the effect of straw mulch on soil evaporation, and the results indicated that straw mulching can significantly reduce soil evaporation, and that mulch resistance is closely related to straw thickness and area index [18,19,20,21,22,23,24]. Moreover, some scholars have also investigated the effect of sand and gravel cover on soil evaporation, which suggested that sand and gravel cover can appreciably reduce soil evaporation [25,26,27,28,29,30,31,32].

Although previous studies put more emphasis on the effects of straw mulch and sand mulch on soil evaporation, there is a scarcity of research on the influence of straw decomposition during the soil evaporation process. Straw decomposition can improve soil structure, thus presenting a positive impact on soil evaporation [33,34,35,36,37]. By increasing soil organic matter and microbial activity [38], returning straw to the field can improve soil erosion resistance and water permeability, thus affecting the dynamic change in soil water. Therefore, straw decomposition plays an important role in regulating soil water evaporation and water retention by improving soil’s physical properties. This paper intends to use the empirical model proposed by Yang and Blackewll et al. [12] to build a soil surface evaporation resistance model under straw decomposition. The resistance model was designed to study the effect of straw decomposition on soil water evaporation.

These are the main objectives of this study: (1) Under the same experimental conditions, establish evaporation resistance models for both straw decomposition and bare soil. By comparing these two models, determine the influence of straw decomposition on soil evaporation. (2) Analyze the underlying causes of straw decomposition’s influence on soil properties using soil water retention curves under different decomposition degrees. 

This study will contribute to an understanding of the changes in soil’s physical properties after straw incorporation and how these changes affect the dynamic balance of soil water. Furthermore, by revealing the mechanism through which straw decomposition impacts soil evaporation, this research provides a scientific basis for farmland water management and guides farmers in adopting more effective moisture retention strategies.

## 2. Results and Analysis

### 2.1. Straw Decomposition Process

The cumulative decomposition rate and average decomposition rate of straw are shown in Figure 1 and Figure 2.

Figure 1 and Figure 2 show the cumulative decomposition rate and decomposition rate of straw of the two experiments. From Figure 1a and Figure 2a, it can be seen that the cumulative decomposition rate curve exhibits a trend of rapid increase in the early stage, followed by a stabilization in the later stage. From Figure 1b and Figure 2b, in the first two stages, the decomposition rate is relatively fast.

As shown in Figure 1a and Figure 2a, the cumulative decomposition rate in the first experiment reached 21.11% during the first two stages, while, in the second experiment, it reached 30.45%. In the subsequent stages of decomposition, the cumulative decomposition rate in the first experiment was 33.86%, and in the second experiment it reached 53.83%. As shown in Figure 1b, in the first experiment, the decomposition rate peaked during the first two stages, with an average decomposition rate of 1.93 g/d in the first stage and 3.35 g/d in the second stage. After that, the decomposition rate began to decline, with the final stage showing an average decomposition rate of 0.48 g/d. As shown in Figure 2b, in the second experiment, the highest decomposition rate occurred in the first stage, at 5.97 g/d, followed by fluctuations in the decomposition rate, with the average decomposition rate in the final stage being only 1.07 g/d.

### 2.2. The Changes in Surface Soil Moisture Content over Time for Two Different Treatments at Each Stage

As illustrated in Figure 3 and Figure 4, the trends in surface soil moisture content (0–1 cm) over time for each treatment in both experiments exhibit a generally similar pattern, characterized by an exponential decrease. This decrease is significantly negatively correlated with time (*p* < 0.01).

Specifically, as shown in Figure 3, during the first stage of the experiment, the decline in soil moisture content for JG1-1 was slower compared to CK1-1. In the second and third stages, JG1-2 and JG1-3 showed a more significant downward trend in soil moisture content compared to the corresponding CK1-2 and CK1-3. However, in the fourth and fifth stages, JG1-4 and JG1-5 exhibited a relatively lower decline in soil moisture content compared to CK1-4 and CK1-5. These findings indicate that, during the early stage of straw decomposition (0–40 days), the decomposition process did not significantly enhance the soil water retention capacity. Rather, in the mid-to-late stages of decomposition (after 40 days), the effect of straw decomposition began to show an enhancement in soil water retention capacity gradually.

The apparent enhancement of water retention during the early stages of decomposition may merely be due to the mixing of straw within the topsoil layer, which hindered the upward movement of water vapor molecules, thereby creating a phenomenon of “apparent” water retention enhancement. The results of the second experiment were consistent with those of the first experiment. Within the first 40 days of straw decomposition, the decomposition process did not significantly improve the soil water retention capacity. However, after 40 days of decomposition, the soil moisture content under the straw decomposition treatment consistently remained higher than that of the bare soil control under the same conditions, and its declining trend was also more moderate.

### 2.3. Variation of Cumulative Evaporation with Soil Moisture Content

Figure 5 and Figure 6 demonstrate the variation of cumulative evaporation with surface soil moisture content (0–1 cm) at each stage of the two experiments. The cumulative evaporation at each stage shows a linear decreasing trend with increasing moisture content, reaching a highly significant level (*p* < 0.01). The coefficient of determination for the fitted lines at each stage surpasses 0.9, indicating a close correlation. Table 1 provides the cumulative evaporation for each experimental stage. In the first experiment, the cumulative evaporation of soil under straw decomposition treatment was consistently lower than that of bare soil treatment across all stages. Similarly, in the second experiment, except for the fourth stage, the cumulative evaporation in the straw decomposition treatment remained lower than that of the bare soil treatment in the other five experimental stages. This indicates that straw decomposition is conducive to suppressing evaporation. In the first experiment, the relative reduction ranged from 3.08% to 32.2%, with the maximum relative reduction of 32.2% observed in the fourth stage (JG1-4). In the second experiment, the relative reduction ranged from 16.02% to 29.3%, with the maximum relative reduction of 29.3% observed in the second stage (JG2-2).

### 2.4. Soil Moisture Retention Curves for Bare Soil and 60 Days’ Decomposition

The soil water retention curves of bare soil and 60 days’ decomposition soil were selected to study the effect of continuous decomposition on soil water retention capacity. 

The fitting results of the four parameters $\theta_{s}{,\theta }_{r},\alpha ,$ n for the soil moisture retention curves for CK1-1 and JG1-5 in the first experiment are presented in Table 2, with the corresponding curves illustrated in Figure 7. The soil moisture retention curve is a crucial indicator of the soil’s ability to retain water, reflecting the relationship between soil water suction and soil moisture content. As depicted in Figure 7, in the first experiment, when the soil water suction is less than 9160 cm H_2_O, the CK1-1 curve is higher than that of JG1-5. When the soil water suction is 4000 cm H_2_O, the corresponding moisture content of CK1-1 is 0.282 higher than that of JG1-5. This indicates that, within this suction range, the soil’s water retention capacity after 60 days of straw decomposition is lower compared to bare soil. However, when the soil water suction exceeds 9160 cm H_2_O, the JG1-5 curve consistently surpasses the CK1-1 curve. At the water suction of 14,000 cm H_2_O, the corresponding moisture content of CK1-1 is 0.018 lower than that of JG1-5, suggesting that, in this higher suction range, the soil’s water retention capacity after 60 days of straw decomposition is superior to bare soil.

In the second experiment, the fitting results of parameters A and B for the soil moisture retention curves of JG2-6 and CK2-1 are presented in Table 3, and the curves are illustrated in Figure 8. According to Figure 8, the JG2-6 curve is consistently higher than that of CK2-1 across the entire suction range. When the soil water suction is 4000 cm H_2_O, the corresponding moisture content of CK2-1 is 0.021 lower than that of JG2-6. It indicates that the water retention capacity of the soil after 60 days of straw decomposition is consistently higher than that of bare soil.

From the analysis of the two experiments, it can be concluded that the water retention capacity of the soil after 60 days of straw decomposition improved compared to the bare soil. This improvement is particularly significant in the medium and high suction ranges. Therefore, it can be inferred that the straw decomposition process contributes to the enhancement of soil water retention performance.

### 2.5. Construction of Soil Surface Evaporation Resistance Model for Each Treatment and Stage

The soil surface evaporation resistance model curves for the two treatments at each stage of the first experiment are depicted in Figure 9. The model is fitted according to Equation (12), and the varying trends of surface evaporation resistance with moisture content at each stage are highly significant (*p* < 0.01). The regression results for parameters a, b, and c are presented in Table 4.

From Figure 9a, it can be seen that, in the first stage, as evaporation progresses, the moisture content continuously decreases. Specifically, when the moisture content is above 23.4%, the soil surface evaporation resistance of JG1-1 exceeds that of CK1-1; when the moisture content is below 23.4%, the surface evaporation resistance of JG1-1 is lower than that of CK1-1. Figure 9b shows that, in the second stage, the surface evaporation resistance of JG1-2 remains consistently higher than that of CK1-2 throughout the evaporation process. Figure 9c reveals a similarity between the third and first stages: when the moisture content is above 20.8%, the surface evaporation resistance of JG1-3 is greater than that of CK1-3; when the moisture content is below 20.8%, the situation is reversed. Figure 9d indicates that, in the fourth stage, the surface evaporation resistance of JG1-4 is consistently higher than that of CK1-4 throughout the evaporation process. Figure 9e shows that, in the fifth stage, the surface evaporation resistance of JG1-5 is significantly greater than that of CK1-5 throughout the evaporation process.

The model curves illustrating soil surface evaporation resistance for the two treatments at each stage of the second experiment are shown in Figure 10, with the regression results for model parameters presented in Table 5. Analyzing Figure 10a reveals that, in the first stage, JG2-1 consistently demonstrates higher soil surface evaporation resistance compared to CK2-1 throughout the evaporation process. Figure 10b indicates that the surface evaporation resistance of JG2-2 is greater than CK2-2 when soil moisture content exceeds 9.5%, but it is lower when soil moisture content falls below 9.5%. Figure 10c indicates that soil surface evaporation resistance of JG2-3 is lower than CK2-3 when the moisture content surpasses 21.95%, but this reverses when the soil moisture content is below this threshold. Figure 10d demonstrates that the soil surface evaporation resistance of JG2-4 is greater than CK2-4 when soil moisture content exceeds 14.5%, but it is less when the soil moisture content falls below 14.5%. As shown in Figure 10e, f, in the fifth and sixth stages, both JG2-5 and JG2-6 consistently exhibit higher surface evaporation resistance compared to CK2-5 and CK2-6, respectively.

In the first experiment, the values of coefficient R^2^ are between 0.521 and 0.804. In the second experiment, the values of coefficient R^2^ are between 0.374 and 0.744. They reveal a less than good fit between soil surface evaporation resistance and soil moisture content. The reason is that there are errors in the actual measurements. As evaporation progresses, there will be some errors in the evaporation amount measured by the lysimeters.

In summary, the analysis results from both experiments indicate that straw decomposition enhances soil surface evaporation resistance, and this enhancement becomes more pronounced as decomposition progresses.

### 2.6. Model Simulation Accuracy

The predicted evaporation and the measured evaporation for each stage of the first experiment are depicted in Figure 11, while the evaluation parameter calculation values are provided in Table 6. The Nash–Sutcliffe efficiency coefficient (NS) ranges from negative infinity to 1, with values closer to 1 indicating better model quality and higher reliability. Conversely, the Sum of Squares Error (SSE), Root Mean Square Error (RMSE), and Mean Absolute Error (MAE) values closer to 0 signify higher model accuracy. As shown in Table 6, the R^2^ values of JG1-4 exceed 0.7, indicating that the model of JG1-4 is highly accurate. The other R^2^ values range between 0.5 and 0.7, indicating that the models have medium accuracy. The NS coefficients for all stages are all greater than 0, confirming the model’s reliability. Notably, the model accuracy in JG1-4 is relatively high, with an NS coefficient of 0.9063 and an R^2^ value of 0.926, suggesting an excellent match between the predicted and measured evaporation values in this stage.

The models for the other four stages do not have high accuracy, possibly due to measurement errors in the measured evaporation intensity during the experiment.

The predicted and measured evaporations for each stage of the second experiment are shown in Figure 12, and the evaluation parameter calculation values are presented in Table 7. As indicated in Table 7, the NS coefficient values for all six stages of the second experiment are greater than 0. The SSE, RMSE, and MAE coefficients are all close to 0. The minimum value of R^2^ is 0.3824, and the maximum value is 0.6809. It indicates that the models of the second experiment have medium-to-low accuracy.

According to the analysis results of 3.6, the models’ accuracy in the two experiments are at a medium-to-low level, with only JG1-4 exhibiting higher accuracy. The reason may be that the evaporation amounts measured by the lysimeters do not equal the actual soil evaporation, and there are measurement errors involved. In addition, during the experiment, measurements of meteorological data, soil moisture content, and the other parameters may also have measurement errors.

## 3. Materials and Methods

### 3.1. Site Description

The experiment was conducted at the Xinxiang Comprehensive Experimental Base of the Chinese Academy of Agricultural Sciences in Henan Province, China (35°14′ N, 113°76′ E), during two distinct growing seasons: the winter wheat season from November 2023 to January 2024 and the summer corn season from July to August 2024. Each experiment lasted for a period of 60 days. The geographical location of the experimental base is shown in Figure 13. The altitude of the experimental base is 81 m, with an average annual temperature of 14 °C, belonging to a warm temperate continental monsoon climate. The frost-free period spans 210 days annually, accompanied by an annual sunshine duration of 2399 h. The area receives an average annual rainfall of 582 mm and experiences an average annual evaporation of 1908.7 mm. Moreover, the experimental soil belongs to the type of sandy loam, possessing a bulk density of 1.32 g/cm^3^ [39,40].

### 3.2. Setup of Experiment

#### 3.2.1. Location of Experiment

The purpose of this experiment was to investigate the straw decomposition process and its subsequent impact on soil moisture. The experiment was conducted in two leaching ponds, each consisting of a concrete pool measuring 3 m in length, 2.4 m in width, and 2 m in depth. To prevent any interference from rainfall, electric rain shelters were installed around the leaching ponds.

#### 3.2.2. Design of Experiment

Leaching pond-1: The upper half was embedded with 18 nylon mesh bags (40 × 60 cm, 200 mesh) filled with broken straw. They were used to quantify the degree of straw decomposition. The lower half consisted of bare soil, serving as a blank control, as shown in Figure 14a. Leaching pond-2: The corn straw was returned to the soil, allowing it to be thoroughly mixed with the tillage-layer soil, as shown in Figure 14b. 

The experiment was conducted in two simultaneous groups: one was the bare soil control treatment (Treatment 1, the lower half of Leaching pond-1); while the other was the straw decomposition treatment (Treatment 2, the upper half of Leaching pond-1, and Leaching pond-2). The first experiment started on 16 November 2023 and ended on 15 January 2024, while the second experiment began on 15 June 2024 and concluded on 14 August 2024. 

First, the corn straw was dried and crushed to a length of 3–5 cm in preparation for later use. The average yield of corn straw in Henan Province stands at approximately 4800 kg per acre [41], and calculations have indicated that each nylon mesh bag in Leaching pond-1 should contain 220 g of straw, which would be buried about 5 cm below the soil layer. For Leaching pond-2, with a weight of 8 kg, the straw should be evenly incorporated back into the field, ensuring that the straw was thoroughly mixed with the tilled soil. 

Next, the period required for the soil moisture content to decrease from the saturation to the wilting point after each irrigation was defined as an experimental stage. When the soil moisture content dropped to the wilting point, the measurement work was concluded. So, the measurement work generally lasted for 6 to 8 days; 10 days were set as an experimental stage in order to facilitate the progress of the experiment and the analysis of data. Therefore, the first experiment was structured into five stages, with the first, second, fourth, and fifth stages spanning 10 days each. However, due to the lower temperature in the third stage, which resulted in a slower evaporation rate, this particular stage was extended to 20 days. The second experiment was divided into six stages, each lasting 10 days. The different treatments for each experimental stage in both experiments are detailed in Table 8 and Table 9.

On the morning of the first day of each experimental stage, both leaching ponds were uniformly irrigated with a volume of 0.2 m^3^ before 7 a.m., which aimed to ensure that the soil in the cultivated layer reached saturation. After the irrigation was completed, a layer of plastic film was immediately placed over the surface of the leaching ponds for 48 h. This step was crucial to ensure that the moisture evenly and thoroughly permeated the soil. Once the moisture infiltration process was complete, lysimeters were installed for subsequent measurements. The installation positions of the lysimeters are shown in Figure 15b.

The lysimeter consists of two parts: an inner bucket and an outer bucket. The inner bucket features an inner diameter of 10 cm, while the outer bucket has an inner diameter of 11 cm. Both buckets share a height of 13 cm. Once the inner bucket was inserted into the soil, it formed a column of undisturbed soil in accordance with the surrounding environment. Subsequently, the outer bucket was placed in the hole from which the inner bucket’s soil was initially taken. The bottom of the inner bucket was sealed with gauze before being placed inside the outer bucket. The top of the inner bucket was equipped with a handle, as depicted in Figure 15a. During the installation process, it was necessary to ensure that the soil level in the lysimeters aligned with the field surface soil level. Additionally, the soil within the lysimeter was replaced every two days, with the change completed by 7 a.m. to guarantee that the test observations could commence promptly at that time.

### 3.3. Observation Items

Regular measurements of a series of environmental and soil parameters took place at 7 a.m. on the third day of each experimental stage, completed at the end of each experimental stage. The specific items to be measured included the following:

(1) Temperature of the soil surface (Ts): this was directly measured using an infrared thermometer. (2) Relative air humidity (RH) and air temperature (Ta): measurements were taken 10 cm above the soil using an air temperature and humidity meter. (3) Average wind velocity (u): measurements were taken using an anemometer. (4) Soil moisture content: this was determined by the drying method. (5) The weight of lysimeters: an electronic scale was used for weight measurement. (6) Soil water potential: this was measured using the WP4C dewpoint potentiometer.

The following are the arrangements for measurement frequency:

(1) On the 3rd and 4th days, measurements were taken every 2 h from 7:00 to 23:00 each day. (2) For the other days of each stage, measurements were taken once every 4 h from 7:00 to 23:00 each day.

The handling of nylon mesh bags was as follows:

(2) On the mornings of the 10th, 20th, 30th, 40th, 50th, and 60th days of the experiment, the nylon mesh bags were taken out, with three replicate samples taken each time. The samples were washed with water and then placed in an oven to dry at 75 °C for 8 h until a constant weight is reached. Then, the dry weight was measured.

### 3.4. Data Processing

#### 3.4.1. Decomposition Data Processing

The calculation methods for cumulative decomposition rate and average decomposition rate are presented as follows: (1)$$m=m_{0}-m_{t}$$(2)v=(m0−mt)t(3)η=[(m0−mt)m0]×100%
where *m* is the cumulative decomposition amount (g); *v* is the decomposition rate (g/d); η is the cumulative decomposition rate (%); *m*_0_ is the initial dry mass of the straw; and *m_t_* is the dry mass of the straw after *t* days of decomposition.

After obtaining the cumulative decomposition rates and average decomposition rates at different times, data processing was conducted using Excel 2016, and graphs were created with Origin 2022.

#### 3.4.2. Calculation Method of Soil Water Retention Curve

In the first experiment, as shown in Equation (4), the soil moisture retention curve takes advantage of the VG model that is applicable to the vast majority of soils and has high predictive accuracy, which contributes to its smoothness and continuity across the entire suction range:(4)$$\theta =\theta_{r}+\frac{\left(\theta_{s}-\theta_{r}\right)}{{\left[1+{\left(\alpha h\right)}^{n}\right]}^{m}}$$
where *θ* is the soil volumetric water content; *h* is the pressure head; *θ_r_* is the residual volume moisture content of soil; *θ_s_* is the saturated volumetric moisture content of soil; and *n* and *m* are parameters related to pore shape and distribution, m=1−1/n. In accordance with the conventions of current soil moisture determination methods, this paper uses soil water suction values (+) to replace pressure head (−), and replaces volumetric water content (cm^3^/cm^3^) with mass water content (g/g) to calculate model parameters.

In the second experiment, the Garden model is applied into the demonstration of the soil moisture retention curve, as shown in Equation 5:(5)$$\theta =A\times h^{-B}$$
where *A* represents the soil water holding capacity and *B* (where 1 *≥ B ≥* 0) indicates the slope of the curve.

Undisturbed soil samples were taken and dried at the end of each stage. Then, water was poured to the dry soil sequentially with a pipette, and the potential soil water was measured at different moisture contents with a WP4C dewpoint potentiometer. The parameters of the model were obtained through nonlinear regression with the ‘nlinfit’ function in MATLAB 2019a, while curve plots were created with MATLAB 2019a. By comparing the soil moisture retention curves of different treatments at each stage, the straw decomposition’s impact on soil’s physical properties can be studied.

#### 3.4.3. Construction of the Soil Surface Evaporation Resistance Model

Referring to the studies by Van Bavel and Hillel and Yang et al. [12,15], the formulas for soil surface evaporation intensity are as follows:(6)$$E=60\frac{q_{vs}-q_{va}}{r_{t}}\;$$(7)$$q_{vs=}\frac{1.323e^{\frac{17.27T_{s}}{T_{s+237.3}}}}{T_{s}+273.16}e^{\frac{\varphi g}{R\left(T_{s}+273.16\right)}}$$(8)$$q_{va}=\frac{2.185\times 0.6018}{273.16+T_{a}}\frac{R_{H}}{100}e^{\frac{17.27T_{a}}{T_{a+237.3}}}\;$$(9)$$r_{t}=r_{a+}r_{s}\;$$(10)$$r_{a}=\frac{1}{uK^{2}}{ln}^{2}(\frac{Z}{Z_{0}})$$
where *E* denotes the soil evaporation rate, measured in mm/min; *q_vs_* represents the soil surface air humidity, in kg/m^3^; *q_va_* is the air humidity, also in kg/m^3^; *r_t_* is the total evaporation resistance of the soil, s/m; *r_a_* is the aerodynamic resistance, s/m; *r_s_* is the soil surface evaporation resistance, s/m; *T_s_* is the soil surface temperature, °C; *R* is the universal gas constant, J/(mol·K), taken as 8.314; *φ* is the soil surface water potential, m; *g* is the acceleration due to gravity, m/s^2^; *T_a_* is the air temperature, in °C; *RH* is the air relative humidity, %; *Z* is the reference height for measuring wind speed, m; *Z_0_* is the roughness length of the surface, m, taken as 0.02 [42]; *u* is the average wind speed at the reference height, m/s; and *K* is the Karman constant, taken as 0.41.

Based on the measured meteorological data and soil-related data, Formulas (6) and (9) are utilized to derive Equation (11), which allows for the calculation of the soil surface evaporation resistance.(11)$$r_{s}=60\times \frac{q_{vs}-q_{va}}{E}-r_{a}$$

A nonlinear regression was performed with the soil moisture content *θ* to obtain an empirical formula that meets practical applications. As the surface soil gradually dries out, the moisture content decreases, following the increase in surface evaporation resistance. Therefore, a soil surface evaporation resistance model can be established [12], as shown in Equation (12):(12)$$r_{s}=a{\times \left(\frac{\theta_{S}}{\theta }\right)}^{b}+c$$
where *θ* is the average moisture content of the surface layer from 0 cm to 1 cm; *a*, *b*, *c* are the parameters fitted according to the measured *r_s_* and *θ*, and *θ_s_* is the saturated moisture content of the soil surface.

#### 3.4.4. Model Accuracy Evaluation

The degree of agreement between the predicted evaporation and the measured evaporation should be evaluated, and there are different formulas applied to the assessment of the model’s accuracy, including Sum of Squares Error (SSE), Nash Efficiency Coefficient (NS), Root Mean Square Error (RMSE), Mean Absolute Error (MAE), and Coefficient of Determination (R^2^):(13)$$SSE=\sum_{i=1}^{n}{\left(P_{i}-O_{i}\right)}^{2}.$$(14)$$NS=1-\frac{\sum_{i=1}^{n}{\left(P_{i}-O_{i}\right)}^{2}}{\sum_{i=1}^{n}{\left(P_{i}-O_{ave}\right)}^{2}}$$(15)$$RMSE=\sqrt{\frac{1}{n}\sum_{i=1}^{n}{\left(P_{i}-O_{i}\right)}^{2}}.$$(16)$$MAE=\frac{1}{n}\sum_{i=1}^{n}\left|P_{i}-O_{i}\right|$$where *P_i_* is the *i*-th simulated value; *O_i_* is the *i*-th measured value; *O_ave_* is the average measured value; and *n* is the number of simulated or measured values.

## 4. Discussion

### 4.1. Effect of Straw Decomposition on Soil Evaporation Process

According to the analysis results in Section 3.2, during the first 40 days of decomposition, the decreased rate of soil moisture content under straw decomposition was higher than that in bare soil in some experimental stages. It indicated that straw decomposition accelerated the soil evaporation during the first 40 days of decomposition. After 40 days of decomposition, the decreased rate of soil moisture content under the straw decomposition was always lower than that in bare soil. It indicated that straw decomposition will reduce the soil evaporation after a certain degree of decomposition. The reason for the acceleration of soil evaporation from 0 to 40 days after returning straw to the field may be that the broken straw mixed into the surface soil formed large soil pores. Then, the water retention capacity of soil declined; thus, the soil evaporation increased. At 40d–60d of decomposition, the decomposition of straw was more complete, the pore structure of soil was well developed, and the water retention capacity of soil was stronger than that of bare soil.

According to the analysis results in Section 3.3, straw decomposition can reduce the cumulative soil evaporation. The reason may be that the broken dry straw absorbed part of water in the soil surface, and the straw was located in the ascending path of water vapor molecules, which hindered the escape of water vapor molecules, thus leading to the decrease in cumulative evaporation.

### 4.2. Effect of Straw Decomposition on Soil Water Retention

According to the analysis results in Section 3.4, straw decomposition contributes to the enhancement of soil water retention capacity, especially in the medium and high suction ranges. 

The decomposition of straw can improve the soil pore structure [43,44]. The straw decomposes by the action of microorganisms, and the humus produced by decomposition changed the sorting of soil pores. After 60 days of decomposition, the macropores in the soil disappeared and the soil structure was perfect. Furthermore, the soil water retention capacity was improved.

According to the analysis results in Section 3.2, during the first 40 days of straw decomposition, in most cases, the water retention capacity of soil with straw decomposition was lower than that of bare soil. It was only after 40 days of decomposition that the soil water retention capacity began to increase. In the early stage of straw decomposition (0–40 days), the decomposition process consumed substantial moisture within the soil. Moreover, the incorporation of straw disrupted the original soil pore structure, creating larger voids that facilitated water loss. Additionally, during straw decomposition, larger diameter capillaries with lower water suction came into being. Therefore, the soil’s ability to retain water decreased, resulting in a general decline in soil moisture. As time progressed, the fragmented straw gradually decomposed and decreased in volume under the action of soil microorganisms, causing soil voids to diminish. At the same time, soil aggregates gradually formed and developed, improving the soil pore structure and enhancing its water retention capacity. The longer the duration of straw incorporation, the more beneficial it was for improving soil structure and augmenting the infiltration capacity of deep soil.

### 4.3. Effect of Straw Decomposition on Soil Surface Evaporation Resistance

With measured data, this study made calculations of the soil surface evaporation resistance, which were then correlated with soil moisture content to derive some empirical formulas for practical applications. Additionally, the decomposition process of straw can be divided into several experimental stages chronologically. Through experimental methods, each experimental stage employs a model of soil surface evaporation resistance under straw decomposition.

An analysis of the surface evaporation resistance models for two different treatments at each stage revealed that straw decomposition significantly enhances surface evaporation resistance, with a more pronounced increase observed as decomposition progresses. One possible reason is that the fragmented straw hinders the upward movement of water vapor in the soil, while the other lies in the fact that straw decomposition alters the soil structure. Numerous research findings have shown that straw decomposition can enhance soil microbial activity and community diversity, contributing to the formation of stable soil aggregates during the decomposition process. Straw decomposition promotes the transformation of exogenous organic matter into soil organic matter, thereby changing the soil structure [45,46]. In summary, both reasons converge on the increased resistance to the upward movement of water vapor within the soil.

## 5. Conclusions

This study investigates the soil evaporation process under straw decomposition, and employs measured data to construct models of soil surface evaporation resistance at different stages of decomposition. The main conclusions are presented below.

(1)The decomposition of straw can reduce soil evaporation. Under identical experimental conditions, the cumulative evaporation measured by the lysimeters during straw decomposition was consistently lower than that observed in the bare soil, with a maximum relative reduction of up to 32.2%.(2)By analyzing the soil moisture retention curves of soil with straw decomposition and bare soil after 60 days, it turned out that straw decomposition could enhance the soil’s water retention capacity. At the same soil water suction, the corresponding moisture content increased by approximately 2.1%.(3)The decomposition of straw increases soil surface evaporation resistance, and the higher the degree of decomposition, the greater the surface evaporation resistance.(4)This study is helpful for the research of farmland water conservation measures.

## Figures and Tables

**Figure 1 plants-14-00434-f001:**
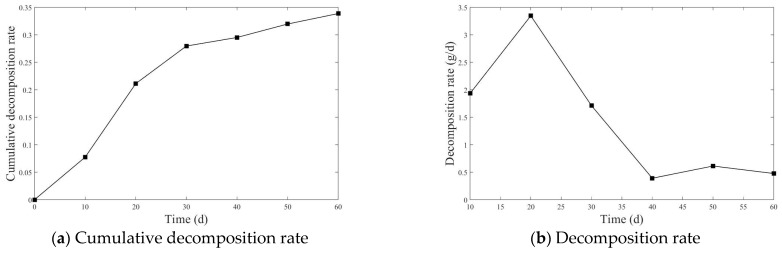
Cumulative decomposition rate and decomposition rate of the first experiment.

**Figure 2 plants-14-00434-f002:**
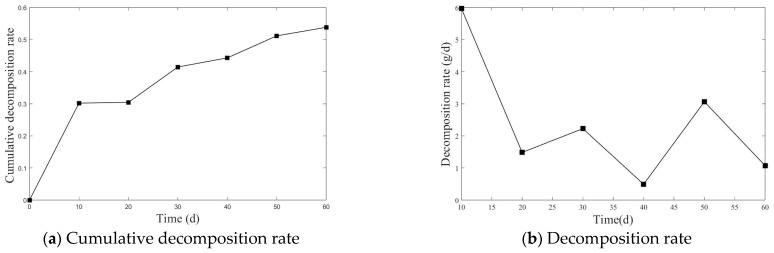
Cumulative decomposition rate and decomposition rate of the second experiment.

**Figure 3 plants-14-00434-f003:**
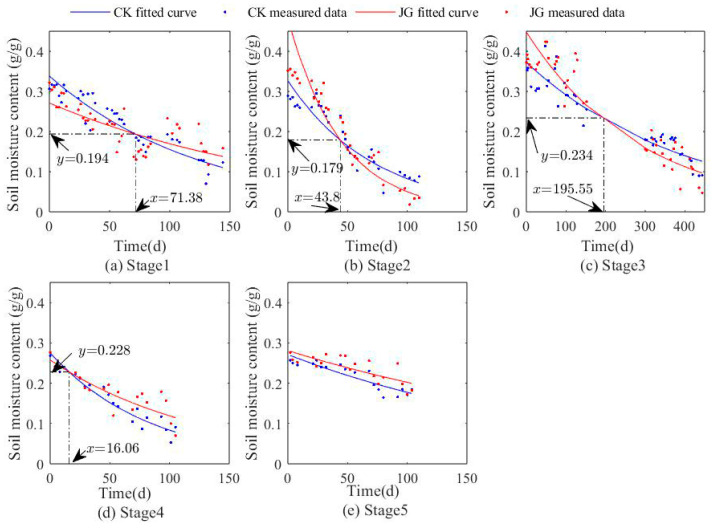
The relationship between surface soil moisture content (0~1 cm) and time in each stage of the first experiment.

**Figure 4 plants-14-00434-f004:**
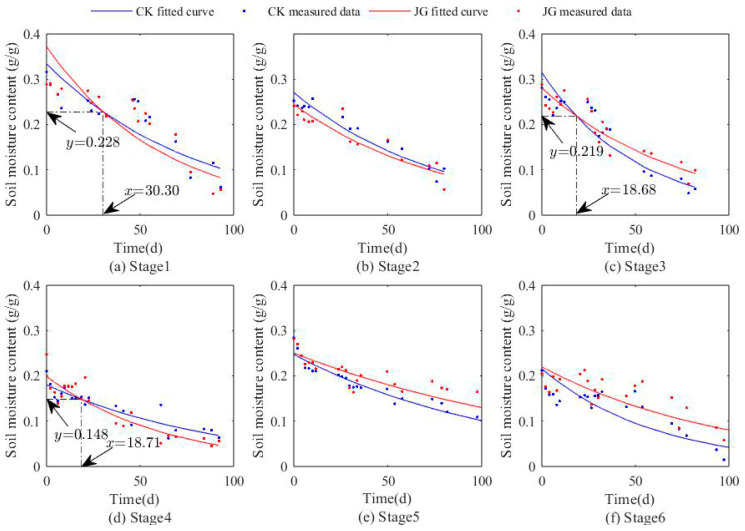
The relationship between surface soil moisture content (0~1 cm) and time in each stage of the second experiment.

**Figure 5 plants-14-00434-f005:**
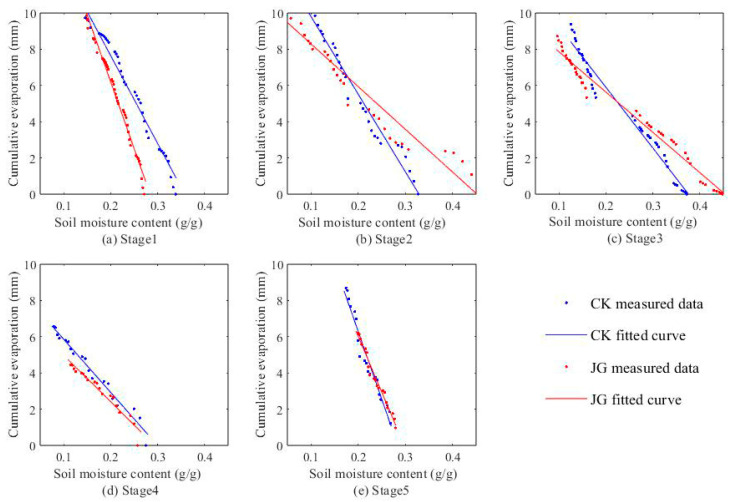
The relationship between cumulative evaporation and surface soil moisture content (0~1 cm) in the first experiment.

**Figure 6 plants-14-00434-f006:**
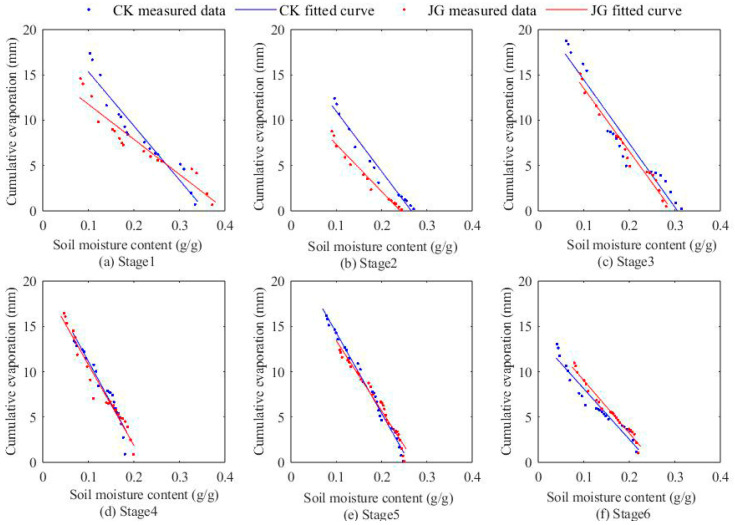
The relationship between cumulative evaporation and surface soil moisture content (0~1 cm) in the second experiment.

**Figure 7 plants-14-00434-f007:**
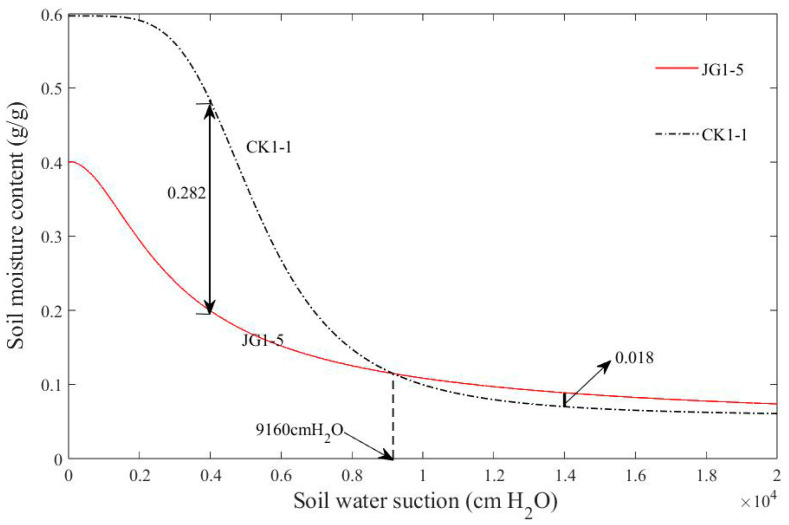
Soil moisture retention curves for each stage of the first experiment.

**Figure 8 plants-14-00434-f008:**
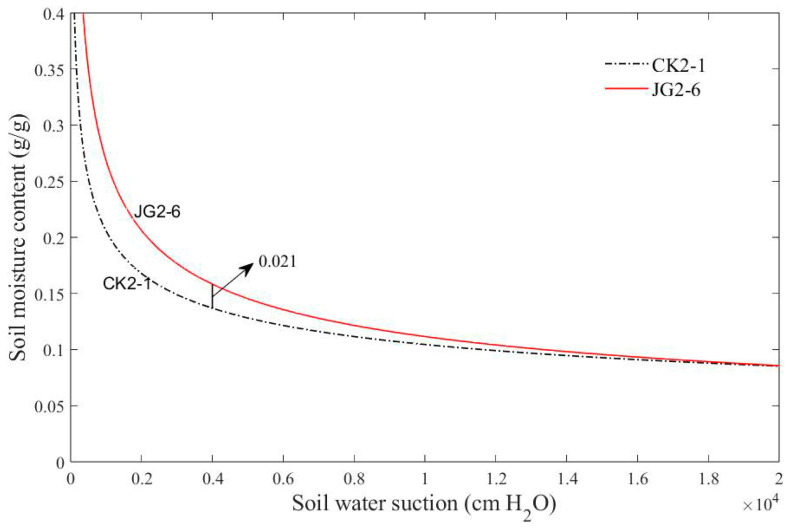
Soil moisture retention curves for the second experiment.

**Figure 9 plants-14-00434-f009:**
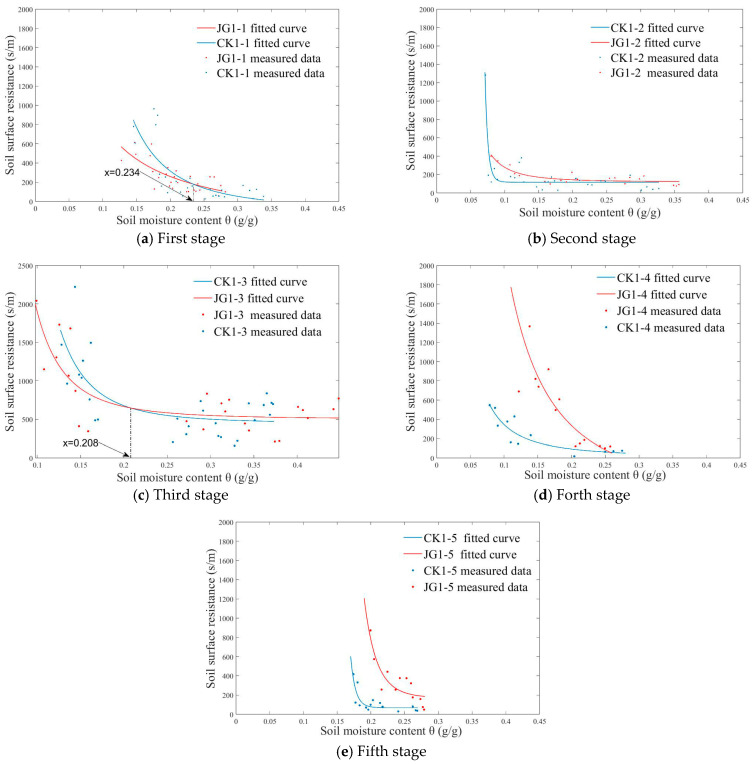
Relationship between soil surface resistance $r_{s}$ and soil moisture content θ in the first experiment.

**Figure 10 plants-14-00434-f010:**
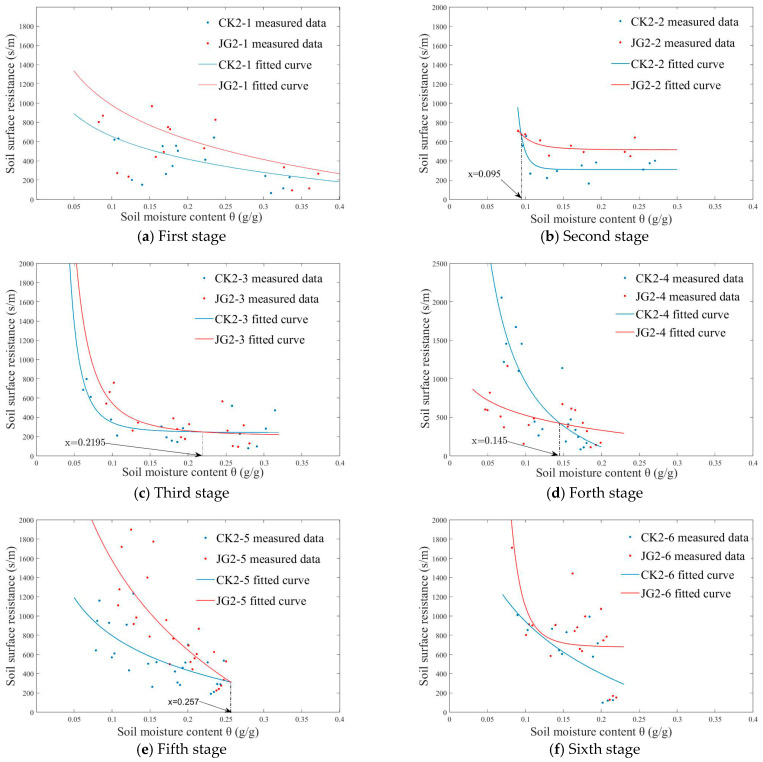
Relationship between soil surface resistance $r_{s}$ and soil moisture content θ in the second experiment.

**Figure 11 plants-14-00434-f011:**
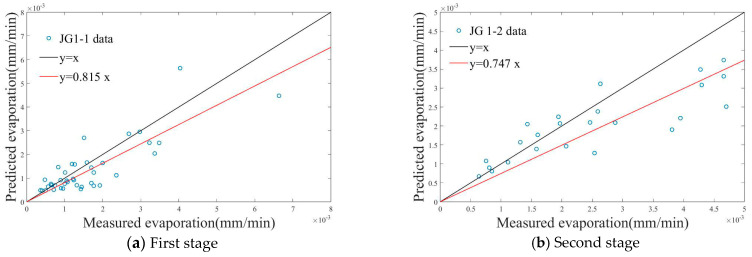
Relationship between predicted evaporation and measured evaporation in the first experiment.

**Figure 12 plants-14-00434-f012:**
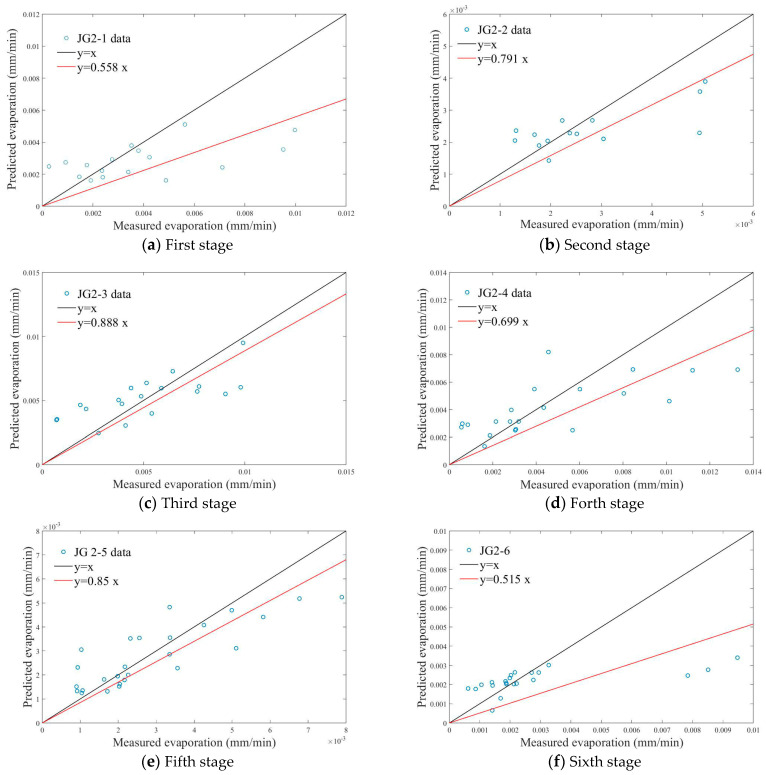
Relationship between predicted evaporation and measured evaporation in the second experiment.

**Figure 13 plants-14-00434-f013:**
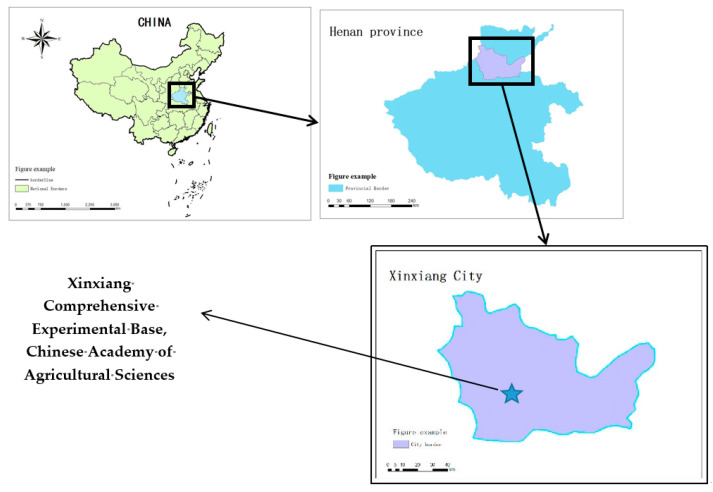
The location of Xinxiang comprehensive experimental base. Note: The experimental base is located in Xinxiang City, Henan Province, China.

**Figure 14 plants-14-00434-f014:**
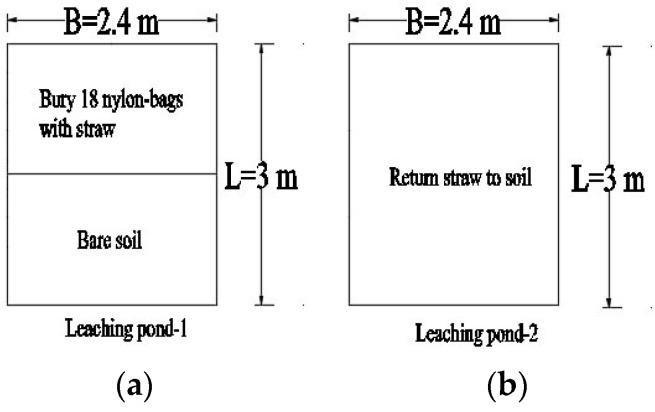
Layout of two leaching ponds. a: 18 nylon bags were buried in half of Leaching pond-1, the other half was totally bare soil. Nylon bags were filled with straw in order to measure the decomposition rate. b: returning straw to the field and evenly mixing the straw with the surface soil.

**Figure 15 plants-14-00434-f015:**
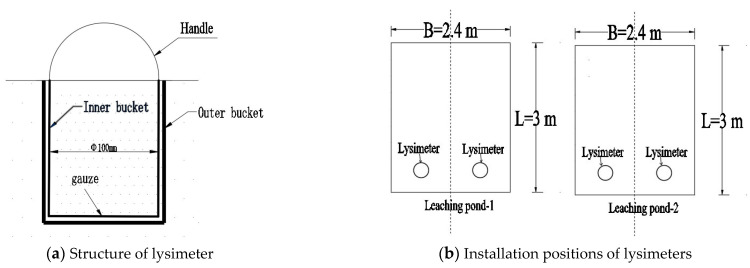
Structure and layout diagrams of lysimeter.

**Table 1 plants-14-00434-t001:** Cumulative evaporation amount at each experimental stage of the two experiments.

Stage Name	Cumulative Evaporation (mm)	Stage Name	Cumulative Evaporation (mm)	Relative Reduction
CK1-1	11.766	JG1-1	10.583	10.06%
CK1-2	11.238	JG1-2	10.893	3.08%
CK1-3	9.390	JG1-3	8.717	7.16%
CK1-4	6.571	JG1-4	4.455	32.20%
CK1-5	8.698	JG1-5	6.208	28.63%
CK2-1	17.380	JG2-1	14.590	16.05%
CK2-2	12.408	JG2-2	8.772	29.30%
CK2-3	18.733	JG2-3	15.138	19.19%
CK2-4	13.383	JG2-4	16.469	
CK2-5	16.162	JG2-5	12.395	23.32%
CK2-6	13.061	JG2-6	10.970	16.02%

**Table 2 plants-14-00434-t002:** Parameter values of soil moisture retention curves for each stage of the first experiment.

Stage Name	$\theta_{s}$	$\theta_{r}$	α	n	$R^{2}$
CK1-1	0.5972	0.0571	0.0002	4.6213	0.981
JG1-5	0.4008	0.0381	0.000521	2.0068	0.943

**Table 3 plants-14-00434-t003:** Parameter values of soil moisture retention curves for each stage of the second experiment.

Stage Name	A	B	R^2^
CK2-1	1.567	0.294	0.971
JG2-6	3.729	0.381	0.852

**Table 4 plants-14-00434-t004:** Fitting results of evaporation resistance model parameters for each stage of the two treatments in the first experiment.

Stage Name	a	b	c	R^2^
CK1-1	22.873	2.621	−86.013	0.591
JG1-1	221.378	0.996	−284.515	0.547
CK1-2	1.893 × 10^−13^	17.084	115.942	0.772
JG1-2	1.410	2.778	119.437	0.791
CK1-3	4.098	3.673	449.939	0.521
JG1-3	9.058	3.110	503.812	0.636
CK1-4	10.092	1.968	5.385	0.804
JG1-4	233.708	1.756	−441.722	0.770
CK1-5	2.69 × 10^−11^	24.370	67.368	0.654
JG1-5	0.403	10.524	167.798	0.756

a, b, c are model parameters.

**Table 5 plants-14-00434-t005:** Fitting results of evaporation resistance model parameters for each stage of the two treatments in the second experiment.

Stage Name	a	b	c	R^2^
CK2-1	1.70212 × 10^5^	0.002	−1.70034 × 10^5^	0.440
JG2-1	5.13754 × 10^5^	0.001	−5.13463 × 10^5^	0.374
CK2-2	1.661 × 10^−5^	11.641	310.551	0.483
JG2-2	0.497	4.962	516.169	0.601
CK2-3	0.874	3.429	239.120	0.651
JG2-3	10.698	2.595	201.868	0.550
CK2-4	317.565	1.132	−589.328	0.744
JG2-4	0.93664 × 10^5^	0.003	−0.93523 × 10^5^	0.461
CK2-5	3410.227	0.133	−3310.380	0.467
JG2-5	1.10831 × 10^5^	0.012	1.11145 × 10^5^	0.634
CK2-6	1.95654 × 10^5^	0.004	1.95805 × 10^5^	0.483
JG2-6	0.136	5.582	675.497	0.521

a, b, c are model parameters.

**Table 6 plants-14-00434-t006:** Calculation results of model accuracy evaluation parameters for the first experiment.

Stage Name	SSE × 10^−6^	NS	RMSE × 10^−4^	MAE × 10^−4^	R^2^
JG1-1	15.070	0.6417	7.23	5.23	0.683
JG1-2	5.121	0.5416	9.12	6.67	0.558
JG1-3	0.358	0.1321	1.36	1.08	0.534
JG1-4	1.658	0.9063	3.35	2.45	0.926
JG1-5	1.599	0.6573	5.42	3.31	0.625

**Table 7 plants-14-00434-t007:** Calculation results of model accuracy evaluation parameters in the second experiment.

Stage Name	SSE × 10^−5^	NS	RMSE × 10^−4^	MAE × 10^−4^	R^2^
JG2-1	1.143	0.1367	25.0	17.0	0.3824
JG2-2	0.265	0.4048	10.0	7.3	0.4697
JG2-3	2.257	0.5029	19.1	16.0	0.5004
JG2-4	3.537	0.4500	26.0	19.0	0.5092
JG2-5	1.317	0.6704	11.0	8.0	0.6809
JG2-6	0.420	0.1735	22.0	12.0	0.3655

**Table 8 plants-14-00434-t008:** The first experiment: the marking table for each experimental stage of the two different treatments.

	1 (1 d~10 d)	2 (11 d~20 d)	3 (21 d~40 d)	4 (41 d~50 d)	5 (51 d~60 d)
Treatment1	CK1-1	CK1-2	CK1-3	CK1-4	CK1-5
Treatment2	JG1-1	JG1-2	JG1-3	JG1-4	JG1-5

CK—Bare soil (blank control). JG—Returning straw. 1—First stage. 2—Second stage. 3—Third stage. 4—Forth stage. 5—Fifth stage.

**Table 9 plants-14-00434-t009:** The second experiment: the marking table for each experimental stage of the two different treatments.

	1 (1 d~10 d)	2 (11 d~20 d)	3 (21 d~30 d)	4 (31 d~40 d)	5 (41 d~50 d)	6 (51 d~60 d)
Treatment1	CK2-1	CK2-2	CK2-3	CK2-4	CK2-5	CK2-6
Treatment2	JG2-1	JG2-2	JG2-3	JG2-4	JG2-5	JG2-5

CK—Bare soil (blank control). JG—Returning straw. 1—First stage. 2—Second stage. 3—Third stage. 4—Forth stage. 5—Fifth stage. 6—Sixth stage.

## Data Availability

The original contributions presented in the study are included in the article; further inquiries can be directed to the corresponding author.

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
