# Peer review of "Effects of Straw Decomposition on Soil Surface Evaporation Resistance and Evaporation Simulation"

_plants, 2025, doi:10.3390/plants14030434_

Round 1
Reviewer 1 Report
Comments and Suggestions for Authors
Dear Authors,
Your manuscript ‘Effects of Straw Decomposition on Soil Surface Evaporation Resistance and Evaporation Simulation’ is very interesting.
This study investigated the variations in soil evaporation rate, moisture content over 60 days after straw returning to the field and bare soil through two measurement pit experiments.
The main conclusions are: The moisture content of the surface soil decreases exponentially over time, and after 40 days of straw decomposition, the water content of the soil under decomposition is higher than that of bare soil. As the moisture content decreases, the cumulative evaporation from the soil increases linearly. The cumulative evaporation from the decomposed straw soil is lower than that of bare soil, with a relative reduction ranging from 3.08% to 32.2%. The straw decomposition significantly enhanced the water retention capacity of the soil in the medium to high suction range; The straw decomposition enhances the evaporation resistance of the soil surface, and the greater the degree of decomposition, the more significant the enhancement effect.
I believe that the manuscript is of potential interest for readers.
My specific comments, that I hope will help the authors to improve the manuscript:
Abstract:
In my opinion it is well structured, and the most important conclusions of the study are well presented there.
Introduction:
- I would suggest adding more recent references in the introduction. if it is possible, I suggest making the substitution.
- The introduction is well-developed and fits in with the subject matter.
Materials and Methods:
- 2.1 - The location of the tests is shown in Figure 1. The duration of the test is also shown.
- Figure 1 needs a full caption, I suggest adding more information to it.
-2.2 - The design of the experiment is shown in Figure 2. The text explains how the experiment was carried out.
- Figure 2, like figure 1, needs additional information.
- Table 1 and table 2 I suggest reducing the text in the first line. You could, for example, replace them with 1, 2, 3 ... and in the caption say 1-First stage
- Figure 3, caption needs to be rewritten
-2.3 - Well done, looks good
-2.4 - equations are numbered and quoted in the text
Results:
- Models are presented based on the results obtained. The length of the graphs presented may cause the reader to lose focus on the most important results
-at the end of the results section, the models and tables with the statistical analysis are presented
- I suggest restructuring the text so that there is no blank space on page 16.
-The division of the presentation of the results makes it easier to read and understand the results obtained in the study
Discussion:
-The discussion could be improved.
-Many results are presented that could be discussed further in this part
-so many results are presented that one would expect them to be discussed in this section, but this is not the case.
- I suggest discussing the results with recent studies in the field. The bibliography used needs to be improved.
Conclusions:
- The conclusion presents the most important results of the study
-is structured and the main conclusions are understood
References:
It’s ok.
I suggest introducing the DOI's that are missing from some references
Reviewer 2 Report
Comments and Suggestions for Authors
- Citation in the main content should be improved as MDPI style. For example, “Van Bavel and Hillel (1976)” should be “Van Bavel and Hillel [15]”
-Lines 61-62: Rice straw returning increase OM and enhance soil bacteria activates. This statement needs the reference. Please see this paper. [https://doi.org/10.3390/biology12040501]
-Lines 86-91: Please refer to the source of this data.
-Figures 2 and 3: The text are difficult to read. Must be improved.
- There are too many figures. Most of them are small text and poor resolution. The authors must eliminate and improve them.
-Results are many, but discussions are very short. The authors must improve more discussion by following the objectives. For example, water retention capacity and straw decomposition should be discussed. Please see this paper. [https://doi.org/10.3390/su162310588]
- The relationship of evaporation and straw decomposition should be discussed.
- The relationship of straw decomposition, soil properties, and soil water retention should be discussed.
-Conclusion: Recommendation for practicing in the farm should be provided.
Reviewer 3 Report
Comments and Suggestions for Authors
Review article ID - plants-3395705 entitled “Effects of Straw Decomposition on Soil Surface Evaporation Resistance and Evaporation Simulation”
The article deals with a study conducted at an experimental base in Xinxiang (China) that evaluates changes in soil characteristics from the application of straw.
Suggestions for typographical and editorial corrections are indicated in the attached text.
Considerations were made in some passages that aim to contribute to the clarity of the writing.
In general, the article is verbose and repetitive, which should be carefully revised.
The authors should or could better layout the article in order to optimize the space on each page.
The structure of the article is methodologically adequate.
The introduction addresses the topic adequately and with good bibliographic support, but there are problems with the citations. As indicated in the text, cited references do not correspond to the referenced ones, which may reflect a chain problem with the other citations and references. This requires careful review and correction.
The materials and methods are described in such a way as to describe the experiment performed in detail. However, the language is sometimes verbose and repetitive, which should be more direct, making the text more objective for future readers.
The article describes an experimental work performed, but in part of the text they describe it as if it were still to be done (lines 149, 152 and 175). It needs to be revised.
The results and analyses present the data obtained in the form of tables and figures. The size of the figures could be standardized and the tables should adopt a uniform font size.
The discussions are too succinct for the large amount of data produced; they should discuss the accuracy of the results in greater depth, considering the respective limitations of the methods for obtaining accuracy. The results should be better discussed and compared with studies carried out in other countries and continents.
The study, from the title of the article, suggests the intention of producing an article with broad application and, therefore, pairing the study with other studies conducted in other parts of the world will take away the local or regional validity of the article.
The references should be carefully reviewed, as there may be unreferenced citations.
The references do not follow a single standard, nor the one adopted by the journal, regarding the way of naming the authors, nor the way of abbreviating the names of the journals. The names of the journals should be placed in italics. The DOI of all references that have it should be included.

Review article ID - plants-3395705 entitled “Effects of Straw Decomposition on Soil Surface Evaporation Resistance and Evaporation Simulation”
The article deals with a study conducted at an experimental base in Xinxiang (China) that evaluates changes in soil characteristics from the application of straw.
Suggestions for typographical and editorial corrections are indicated in the attached text.
Considerations were made in some passages that aim to contribute to the clarity of the writing.
In general, the article is verbose and repetitive, which should be carefully revised.
The authors should or could better layout the article in order to optimize the space on each page.
The structure of the article is methodologically adequate.
The introduction addresses the topic adequately and with good bibliographic support, but there are problems with the citations. As indicated in the text, cited references do not correspond to the referenced ones, which may reflect a chain problem with the other citations and references. This requires careful review and correction.
The materials and methods are described in such a way as to describe the experiment performed in detail. However, the language is sometimes verbose and repetitive, which should be more direct, making the text more objective for future readers.
The article describes an experimental work performed, but in part of the text they describe it as if it were still to be done (lines 149, 152 and 175). It needs to be revised.
The results and analyses present the data obtained in the form of tables and figures. The size of the figures could be standardized and the tables should adopt a uniform font size.
The discussions are too succinct for the large amount of data produced; they should discuss the accuracy of the results in greater depth, considering the respective limitations of the methods for obtaining accuracy. The results should be better discussed and compared with studies carried out in other countries and continents.
The study, from the title of the article, suggests the intention of producing an article with broad application and, therefore, pairing the study with other studies conducted in other parts of the world will take away the local or regional validity of the article.
The references should be carefully reviewed, as there may be unreferenced citations.
The references do not follow a single standard, nor the one adopted by the journal, regarding the way of naming the authors, nor the way of abbreviating the names of the journals. The names of the journals should be placed in italics. The DOI of all references that have it should be included.
Round 2
Reviewer 2 Report
Comments and Suggestions for Authors
Accept in present form.
Author Response
Dear editor and reviewer,
Thank you for offering us an opportunity to improve the quality of our submitted manuscript (plants-3395705). We appreciated very much you give us constructive and insightful comments.
Sincerely yours,
Shengfeng Wang
Reviewer 3 Report
Comments and Suggestions for Authors
Review version v2 of article ID plants-3395705: “Effects of soil decomposition on soil surface evaporation resistance and evaporation simulation”
Dear editors and authors,
The new version of the article presents a significant improvement over the previous version.
But first, it is worth noting that many of the recommendations and suggestions contained in the comments made in the first review of the article were not accepted or justifiably refuted by the authors.
I suggest that we review the comments made in v1 or, if they disagree, that they justifiably refute them.
The article has typing problems (some indicated in v2). Some citations have errors [Error! Reference source not found.-]
The article requires careful review of the written wording, some suggestions made in v1 are not considered in v2.
The article has a very confusing number of tables, with cited tables that do not correspond to their respective tables and cited tables that do not appear in the text.
References must follow journal standards, as stated in v1, and journal names must be abbreviated.
In view of the above, I am adding two revised versions v1 and v2 and recommend that the article be accepted after major corrections.

Review version v2 of article ID plants-3395705: “Effects of soil decomposition on soil surface evaporation resistance and evaporation simulation”
Dear editors and authors,
The new version of the article presents a significant improvement over the previous version.
But first, it is worth noting that many of the recommendations and suggestions contained in the comments made in the first review of the article were not accepted or justifiably refuted by the authors.
I suggest that we review the comments made in v1 or, if they disagree, that they justifiably refute them.
The article has typing problems (some indicated in v2). Some citations have errors [Error! Reference source not found.-]
The article requires careful review of the written wording, some suggestions made in v1 are not considered in v2.
The article has a very confusing number of tables, with cited tables that do not correspond to their respective tables and cited tables that do not appear in the text.
References must follow journal standards, as stated in v1, and journal names must be abbreviated.
In view of the above, I am attaching revised version v2 and recommend that you consider revised version v1 again.
I recommend that the article be accepted after major corrections.
Round 3
Reviewer 3 Report
Comments and Suggestions for Authors
Review article ID - plants-3395705 titled “Effects of straw decomposition on soil surface evaporation resistance and evaporation simulation” Version 3.
In version 3 of the article, the authors have addressed almost all of the suggested corrections. However, the article still needs a complete rewrite. For example, in line 43 they cite Yang [12] which does not correspond to reference 12.
In line 65 they write “model proposed by Yang (formula 12)” which must also be the result of a typo!
In line 226 it says “Referring to the studies by Van Bavel and Yang et al [12, 15],” which should be corrected to: Referring to the studies by Van Bavel and Hillel and Yang et al. [12, 15],
In view of the above, I recommend that the article be accepted after rewrite, correcting errors such as the errors recorded above.
Comments on the Quality of English LanguageI strongly recommend that it be reviewed by experts in scientific writing in the English language, to ensure that it complies with the quality standards of the Plants journal.
